# Impact of Capsid and Genomic Integrity Tests on Norovirus Extraction Recovery Rates

**DOI:** 10.3390/foods12040826

**Published:** 2023-02-15

**Authors:** Philippe Raymond, Sylvianne Paul, Rebecca A. Guy

**Affiliations:** 1St-Hyacinthe Laboratory—Food Virology, Canadian Food Inspection Agency (CFIA), St-Hyacinthe, QC J2S 8E3, Canada; 2National Microbiology Laboratory, Division of Enteric Diseases, Public Health Agency of Canada (PHAC), Guelph, ON N1G 3W4, Canada

**Keywords:** norovirus, intercalating dye, chelating agent, RNase, capsid integrity, long RT-qPCR, ISO 15216-1:2017

## Abstract

Human norovirus (HuNoV) is the leading pathogen responsible for food-borne illnesses. However, both infectious and non-infectious HuNoV can be detected by RT-qPCR. This study evaluated the efficiency of different capsid integrity treatments coupled with RT-qPCR or a long-range viral RNA (long RT-qPCR) detection to reduce the recovery rates of heat inactivated noroviruses and fragmented RNA. The three capsid treatments evaluated (RNase, the intercalating agent PMAxx and PtCl_4_) reduced the recovery of heat inactivated HuNoV and murine norovirus (MNV) spiked on lettuce, when combined with the ISO 15216-1:2017 extraction protocols. However, PtCl_4_ also reduced non-heat-treated noroviruses recovery as estimated by RT-qPCR. The PMAxx and RNase treatments had a similar effect on MNV only. The most efficient approaches, the RNase and PMAxx treatments, reduced the heat-inactivated HuNoV recovery rates estimated using RT-qPCR by 2 and >3 log, respectively. The long RT-qPCR detection approach also reduced the recovery rates of heat inactivated HuNoV and MNV by 1.0 and 0.5 log, respectively. Since the long-range viral RNA amplification could be applied to verify or confirm RT-qPCR results, it also provides some advantages by reducing the risk of false positive HuNoV results.

## 1. Introduction

Human enteric viruses cause the majority of all foodborne illnesses and include mainly human noroviruses (HuNoV), hepatitis A viruses (HAV) and rotaviruses (RoV). Among these enteric viruses, HuNoV are considered the leading cause of foodborne illnesses and outbreaks [1,2]. HuNoV belong to a genetically diverse group of viruses of the *Caliciviridae* family and are transmitted mainly via the fecal–oral route. HuNoV are small (28–35 nm), non-enveloped, single-stranded and positive-sense RNA viruses. The RNA genome of HuNoV is approximately 7500 nucleotides (nt) in length. The 3′ viral RNA ends with a poly A-tail, while the HuNoV RNA 5′ side is capped covalently by a viral protein (VPg). Based on the capsid sequences, there are ten distinct HuNoV genetic groups [3]. Genogroups I, II, IV, VIII and IX infect humans. Genogroup II is the most prevalent in humans and currently contains 26 genotypes. Between 2016 and 2020, HuNoV genogroup II, genotype 4 (GII.4) was detected in 52% of the norovirus outbreaks [4]. The viral RNA can be detected and quantified by molecular tools such as reverse transcription quantitative polymerase chain reaction (RT-qPCR) [5]. RT-qPCR can detect small viral genome fragments that originate from infectious virus with intact genomes as well as degraded genomes from incomplete virus particles and inactivated viruses.

HuNoV can persist in an infectious state for prolonged periods in the environment, in water and in food [6]. HuNoV are extremely contagious and have a low infectious dose (approximately 18 to 1000 virus particles) [7]. In order to detect trace amounts of HuNoV in food matrices, HuNoV have to be extracted and concentrated, for instance, by ultrafiltration, ultracentrifugation, electrostatic charge, affinity column or polyethylene glycol (PEG) precipitation [8]. A PEG-based approach is used in the reference method (ISO 15216-1:2017) for the concentration of HuNoV before the RNA extraction [5]. The rapid detection of HuNoV from contaminated food allows the implementation of control measures such as food alerts or recalls that can limit the extent of food-related outbreaks. On the other hand, the virus is degraded by heat, ultraviolet light (UV), chlorine products and high pressure processing (HPP) [6]. In this case, the detection of a non-infectious virus by RT-qPCR could be the equivalent of a false-positive result. Indeed, for a food safety agency implementing these alerts, the detection of HuNoV RNA from degraded genomes of incomplete or inactivated viruses increases the risk of initiating recalls of a safe product [6,9].

Current HuNoV cell culture systems are difficult and tedious, and do not allow the detection of infectious HuNoV at levels found in contaminated foods [10]. Research labs have relied on surrogate models that can be grown in cell lines, such as murine norovirus (MNV), Tulane virus (TuV) and feline calicivirus (FCV), to assess the impact of food processing and treatment on infectivity [11]. Several other approaches have been suggested to investigate or differentiate infectious from non-infectious HuNoV extracted from food products, using methodologies that evaluate either the viral capsid or genome integrity, or a combination of these approaches, as a proxy of the virus infectivity [9,12,13,14]. Indeed, while an intact viral capsid is required for virus cell attachment and infection, the capsid also protects the viral RNA genome against degradation. Approaches based on the use of RNase, as well as intercalating and chelating agents, such as monoazide dyes, platinum and palladium compounds, to limit RNA amplification and detection have been tested with various levels of success to assess the capsid integrity [13,14,15,16].

One limitation of the capsid integrity approaches is that they do not eliminate the RT-qPCR signal of fragmented or damaged nucleic acids from viruses whose capsid remains intact, such as with defective viruses or viruses treated with UV light [14]. Molecular approaches to assess the genome integrity include multiple genome-wide amplification [17], purification procedure using the virus poly A tail combined with proximal and distal RT-qPCR [18], amplification of long RT-qPCR of near full-length genomes [19], or a combination of various approaches such as long-range reverse transcription and multiple qPCRs [20,21]. While capsid integrity treatments are applied on the extracted virus when the RNA is encapsidated, genomic RNA integrity tests are applied on the extracted RNA. Combining capsid integrity treatment with a longer genome detection approach may provide more insight on virus integrity.

In this study, using heat-treated noroviruses as a model of degraded non-infectious virus spiked in PBS or on food matrices (lettuce), we compared the recovery reduction efficiency of various capsid integrity approaches combined with or without genomic integrity tests. Providing virus integrity information along with the HuNoV routine testing results could facilitate the public health risk assessment process.

## 2. Materials and Methods

### 2.1. Virus Stocks and Inoculum

Murine norovirus-1 was provided by Dr. H. Virgin from Washington University (St-Louis, MO, USA). MNV was propagated in the RAW 264.7 cell line, as previously described, and the viral lot was titrated by plaque assays [22]. Aliquots of HuNoV GII.4 (CFIA-FVR-020) were obtained as stool specimens from the British Columbia Center for Disease Control. MNV and HuNoV inocula were prepared and viral concentrations, in genomic equivalent per microliters (gEq/µL), were assessed as previously described [23]. To estimate the gEq levels of the inoculum, aliquots of virus mixtures were extracted using a QIAcube and the RNeasy mini Kit (Qiagen, Mississauga, ON, Canada). RNA was eluted in 50 µL RNase-free water and 1 μL of RNasin^TM^ Plus RNase Inhibitor (ThermoFisher, Asheville, NC, USA) was added to the eluate before its storage at −80 °C.

### 2.2. Virus Heat Treatment

The viruses were heat-treated using a Thermomixer C (Eppendorf, Mississauga, ON, Canada) for 5 min at 80 °C and then cooled for at least 10 min at 4 °C, while non-heat-treated controls were kept at 4 °C during the same period.

### 2.3. Matrix Inoculation

Romaine lettuce bought at local stores was subdivided into 25 g subsamples and placed in Whirl-Pak filter bags (VWR, Mont-Royal, QC, Canada) at 4 °C until use. Twelve subsamples were processed in each experiment. The subsamples were evenly spiked as described previously [24]. Briefly, the lettuce subsamples were spiked with 100 µL of non-heat-treated or heat-treated viruses and air-dried for 30 min in a biosafety cabinet. Two 25 g lettuce subsamples spiked with 100 µL PBS pH 7.4 (ThermoFisher, Asheville, NC, USA) were included as negative controls in each batch of twelve extractions.

### 2.4. Virus Extraction from Food Matrices Using ISO 15216-1:2017

The noroviruses were extracted from the spiked lettuce subsamples using the ISO 15216-1:2017 method [5]. The capsid integrity experiments were performed on the 500 µL PBS solubilized polyethylene glycol (PEG 8000) (Sigma-Aldrich, Oakville, ON, Canada) pellet. After the capsid integrity treatments, the NucliSens miniMAG kit (Biomérieux, Montréal, QC, Canada) was used to extract the RNA following the manufacturer’s recommendations. RNA was eluted in 100 µL RNase-free water and 1 μL of RNasin^TM^ Plus RNase Inhibitor (ThermoFisher, Asheville, NC, USA) was added to the eluate before its storage at −80 °C.

### 2.5. RNA Extraction without Food Matrices

Both non-heat-treated and heat-treated viruses in 200 µL PBS were also tested with and without the different capsid integrity treatments in triplicate. Following the treatment, the total RNA was extracted using the RNeasy QIAcube kit. 

### 2.6. Capsid Integrity Treatment

In order to evaluate the noroviruses capsid integrity, three different approaches were tested on the control without food matrices and the viruses eluted in PBS from the solubilized PEG pellet as described above (ISO 15216-1:2017). For each capsid integrity treatment, non-treated controls in PBS were processed in parallel and recovery yield results were pooled together. Both non-heat-treated and heat-treated viruses, with and without the different capsid integrity treatments, were tested using triplicate spiked subsamples and the different conditions repeated at least twice.

The first approach using RNase was adapted from Topping, et al. [25]. The viruses were incubated with 40 U of Rnase ONE^TM^ ribonuclease with 1X reaction buffer (Promega, Madison, WI, USA) for 30 min at 37 °C on the Thermomixer C at 300 rpm. The reaction was stopped with the addition of the RNA extraction kit chaotropic solution. Controls without Rnase and reaction buffer were processed in parallel.

The second approach using PtCl_4_ was adapted from Fraise, et al. [15]. The viruses were incubated for 10 min in 2.5 mM PtCl_4_ (Sigma-Aldrich, Oakville, ON, Canada) at 4 °C on the Thermomixer C at 300 rpm. Controls without PtCl_4_ were processed in parallel.

The third approach using PMAxx was adapted from Randazzo, et al. [26]. The viruses were incubated with 0.5% Triton ×100 (ThermoFisher, Asheville, NC, USA) and 100 µM PMAxx^TM^ (VWR, Mont-Royal, QC, Canada) in 1.5 mL Eppendorf microtubes. The virus solution was agitated by inversion followed by incubation in a Thermomixer C at 300 rpm, for 30 min at room temperature in the dark. These samples were placed on a BLU-V system (Qiagen, Mississauga, ON, Canada) and photoactivated for 15 min at 100% intensity. Photoactivation was performed twice with a 15 min break in the dark between each treatment. In parallel, controls without PMAxx treatment were kept for 30 min at 4 °C in the dark during the experiment. The experiment flow charts are described in Appendix A.

### 2.7. Short RNA Quantification (RT-qPCR)

Both MNV and HuNoV GII.4 were quantified by RT-qPCR using the TaqMan Fast Virus 1–Step Master Mix on the Quantstudio 6 system (ThermoFisher, Asheville, NC, USA) as described in Raymond, et al. [23]. Briefly, the primer and probe sets developed by Baert, et al. [27], Kageyama, et al. [28], Loisy, et al. [29], targeting the ORF1 and ORF2 junction regions, were used to generate HuNoV and MNV amplicons of 86 bp and 108 bp, respectively. The primer and probe sets as well as the RNA transcript standards are described in Appendix A.

### 2.8. Long-Range RNA Quantification (Long RT-qPCR)

Each subsample was tested using both RT-qPCR and long RT-qPCR. The long RT-qPCR protocol was adapted from Raymond, et al. [30]. Briefly, the complementary DNAs (cDNA) were synthesized using the Tx30SxN primer and Maxima H Minus reverse transcriptase (ThermoFisher, Asheville, NC, USA). The reverse transcription was performed in a GeneAmp™ PCR System 9700 thermocycler (ThermoFisher, Asheville, NC, USA) using 5 µM primer, 1 µL of 10 mM dNTP mix, 7 µL RNase free water and 5 µL of the total RNA extract. The synthesized cDNA (~2.5 kb) was amplified and detected using a real-time PCR (qPCR) performed on a QuantStudio 6 with the Taq Platinum PCR kit following the manufacturer’s recommendations (ThermoFisher, Asheville, NC, USA). For each norovirus, a master mix was prepared with the equivalent of 13.4 µL RNase-Free water, 2.5 µL Platinum Taq PCR buffer, 1.5 mM MgCl_2_, 200 µM dNTP, 0.05 µL ROX Dye, 1 U of Platinum Taq DNA polymerase, and 0.4 µM of each primer and 0.25 µM of the probe described for the short RNA quantification. A 5 µL aliquot of the cDNA product was added to 20 µL of the master mix, and after an initial denaturation step at 95 °C for 5 min, the cDNA was amplified with 45 cycles of 15 s at 95 °C, 1 min at 60 °C and 1 min at 72 °C.

### 2.9. Long to Short Viral Genome Fragments Ratio

The proportion of long versus short HuNoV GII.4 and MNV RNA fragments was assessed in a subset of inocula extracted as described above (see Virus Stocks and Inoculum). Using the same qPCR for both the long and short RNA fragments, the ratio allows a quantitative estimation of the RNA fragmentation of the virus inoculum. The RNA extracts were diluted, and the cDNA were synthesized using the Tx30SxN to generate the long fragment (~2.5 kb), and either HuNoV COG2R or the MNV RV-ORF1/ORF2 primer to generate the short fragments (~0.1 kb) in paired samples using the reverse transcription conditions described above (see Long-Range RNA Quantification). The real-time PCR (qPCR) was performed on these cDNA. The ratio of long vs. short HuNoV RNA fragment was estimated using the cycle thresholds (Ct) difference. The ratio = 10^(Δ*Ct*(*ls*)/*m*)^ × 100%, where Δ*C_t(ls)_* = *Ct_long_ − Ct_short_* is the *Ct_long_* value of the viral RNA from the inoculum reverse-transcribed using the Tx30SxN primer, minus the *Ct_short_* value of viral RNA from the inoculum reverse-transcribed using the COG2R or RV-ORF1/ORF2 primers. The slope m was estimated using the virus RNA transcript standard curve. 

### 2.10. Viral RNA Recovery Yields

The impact of heat, capsid treatment and detection on norovirus recovery yields was estimated, either by RT-qPCR or long RT-qPCR, using the cycle thresholds (Ct) variation to the inoculum as the reference level for 100% recovery. The viral RNA recovery yields = 10^(Δ*Ct*/*m*)^ × 100%, where Δ*C_t_* = *Ct_recovered_ − Ct_inoculum_* is the *Ct_recovered_* value of extracted viral RNA from the matrix, or control, minus the *Ct_inoculum_* value of viral RNA extracted from the inoculum, and *m* is the slope of the virus RNA transcript standard curve in RT-qPCR. In the case of the long RT-qPCR, the standard curve was prepared using serial dilutions of the HuNoV or MNV RNA extracted without matrices. Outlier values were identified and omitted using the Tukey procedure incorporated into the MedCalc application (v19.3.1) (MedCalc Software Ltd., Ostend, Belgium). 

A comparison of the relative recoveries yields was carried out in Minitab 17 (Minitab. Inc, State College, PA, USA) by fitting a general linear model with the capsid integrity treatment and detection method as factors, with post hoc Dunnett’s test being used to compare results to those obtained by the short RT-qPCR with the control virus without capsid integrity treatment (PBS).

## 3. Results

### 3.1. Virus Inoculum

The average HuNoV GII.4 and MNV inoculum concentrations were 2.5 × 10^4^ gEq (CI95% 2.2–2.7) (n = 19) and 3.6 × 10^4^ gEq (CI95% 2.9–4.3) (n = 10), respectively. On average, both MNV and HuNoV short RT-qPCR and long RT-qPCR standard curve slopes were −3.4 ± 0.1 and −3.7 ± 0.5, respectively. The average ratios of long to short HuNoV and MNV RNA fragments in the inoculum were estimated at 38 ± 5% and 25 ± 7% (n = 15), respectively. The MNV lot tissue culture infectious dose 50 (TCID_50_) was titrated in culture and calculated at 10^7.1^ TCID_50_/_mL_ (CI95% 10^6.6^–10^7.6^). Based on this result, an equivalence ratio of 203 gEq per TCID_50_ (CI95% 139–297) (n = 7) was calculated when testing the lot by RT-qPCR.

### 3.2. Impact of Capsid Integrity Treatment on Virus Recovery without Matrices

#### 3.2.1. Control PBS without Capsid Integrity Treatment

The HuNoV and MNV inoculum RT-qPCR cycle threshold (Ct) values were lower than the long RT-qPCR Ct values by an average of 3.4 and 2.0 Ct, respectively. The impact of the three different capsid integrity treatments on the non-heated, and heat-treated, HuNoV and MNV inocula recovery rates without food matrix was compared to control in PBS and evaluated using both the short RT-qPCR and the long RT-qPCR (Figure 1, Appendix A). When the HuNoV were heat-treated, and no capsid treatment was applied, the recovery yields detected by the RT-qPCR were still relatively high at 31 ± 17% (n = 36) but were reduced to 9.1 ± 9.6% (n = 36) when estimated using the long RT-qPCR, a difference of 0.5 log (*p* < 0.05). Similar observations were made with heat-treated MNV without food matrix. The MNV recovery yields assessed using the RT-qPCR were still high at 44 ± 10% (n = 30), but were reduced to 15 ± 7% (n = 27) when tested with the long RT-qPCR, a difference of 0.4 log (*p* < 0.05).

#### 3.2.2. PMAxx Capsid Integrity Treatment

When compared to the control in PBS, the non-heat-treated HuNoV recovery yields did not significantly vary following the PMAxx treatment when assessed using the RT-qPCR, while a slight reduction to 76 ± 16% (n = 6) was detected using the long RT-qPCR (*p* < 0.05). However, the PMAxx treatment significantly reduced the non-heat-treated MNV recovery rates, as estimated using RT-qPCR as well as long RT-qPCR to 23 ± 26% (n = 6) and 39 ± 13% (n = 6), respectively (*p* < 0.05). The impact of PMAxx on heat-treated viruses was quite significant. The heat-treated HuNoV and MNV recovery yields estimated by RT-qPCR were 0.06 ± 0.04% (n = 6) and 0.06 ± 0.10% (n = 6), respectively (*p* < 0.05). All PMAxx heat-treated HuNoV and MNV samples were negative when tested using the long RT-qPCR. There was no significant difference between the RT-qPCR and long RT-qPCR heat-treated detection results. 

#### 3.2.3. PtCl_4_ Capsid Integrity Treatment

On the other hand, the addition of the PtCl_4_ capsid treatment significantly reduced the non-heat-treated HuNoV recovery, estimated using both detection approaches to 64% (*p* < 0.05). It also reduced the recovery rate of MNV as estimated by RT-qPCR and long RT-qPCR to 26 ± 16% (n = 21) and 43 ± 27% (n = 18), respectively (*p* < 0.05). Like PMAxx, when the PtCl_4_ capsid integrity treatment was applied to heat-treated HuNoV, the recovery yields detected using the RT-qPCR and the long RT-qPCR were only 0.1 ± 0.2% (n = 12) and 0.4 ± 0.5% (n = 12), respectively. No heat-treated MNV could be detected using the long RT-qPCR.

#### 3.2.4. RNase Capsid Integrity Treatment

In contrast, the RNase treatments did not have an impact on non-heat-treated HuNoV recovery yields. It only slightly reduced the non-heat-treated MNV recovery rates to 60 ± 18% (n = 24) (*p* < 0.05). However, both heat-treated HuNoV and MNV signals were greatly reduced following the RNase treatment. The heat-treated HuNoV recovery rates as estimated by RT-qPCR, and long RT-qPCR, were reduced to 1.0 ± 0.2% (n = 12) and 0.10 ± 0.15% (n = 9), respectively. Meanwhile, the MNV recovery yields as estimated by RT-qPCR, and the long RT-qPCR, were 4 ± 3% (n = 12) and 6 ± 2% (n = 9), respectively (*p* < 0.05). Overall, the long RT-qPCR detection method and the different capsid integrity treatments efficiently reduced the recovery yields of heat-treated viruses without food matrix.

### 3.3. Impact of Capsid Integrity Treatment on Virus Recovery from Lettuce

#### 3.3.1. Control PBS without Capsid Integrity

For the non-heat-treated HuNoV spiked on lettuce, eluted and concentrated with the ISO 15216-1:2017 methodology without capsid integrity treatment, the average recovery yields as estimated by RT-qPCR and long RT-qPCR were similar at 58 ± 14% (n = 12) and 46 ± 12% (n = 12), respectively (Figure 2, Appendix A). When the HuNoV inoculum was heat-treated, the RT-qPCR recovery yields decreased only to 19 ± 7% (n = 17), while they were further reduced by 1 log to 2 ± 2% (n = 17), when tested by long RT-qPCR. The non-heat-treated MNV recovery yields tested by RT-qPCR and long RT-qPCR were 51 ± 17% (n = 12) and 77 ± 34% (n = 12), respectively. The heat-treated MNV RT-qPCR results were still relatively high with a recovery yield at 36 ± 14% (n = 17) and were reduced to 10 ± 10% (n = 17), or by 0.5 log, when tested using the long RT-qPCR. 

#### 3.3.2. PMAxx Capsid Integrity Treatment

Compared to the control in PBS, the PMAxx capsid treatment following the ISO 15216-1:2017 extraction process did not impact the non-heat-treated HuNoV RT-qPCR recovery results, while the recovery estimated using the long RT-qPCR was reduced to 23.7 ± 12.4% (n = 6) (*p* < 0.05). Meanwhile, the capsid treatment reduced the non-heat-treated MNV recovery estimated by both RT-qPCR and long RT-qPCR to 15.1 ± 2.7% (n = 6) and 9.3 ± 4.8% (n = 6), respectively (*p* < 0.05). In contrast, the PMAxx treatment was very efficient at eliminating the recovery of heat-treated viruses. The heat-treated HuNoV RT-qPCR and long RT-qPCR recovery result was null (0/6) and limited to 1 ± 2.5% (1/6), respectively (*p* < 0.05). Similarly, the heat-treated MNV recovery detected using the RT-qPCR and long-Range RT-qPCR was only 0.1 ± 0.1% and 0.2 ± 0.4% (n = 6) respectively (*p* < 0.05).

#### 3.3.3. PtCl_4_ Capsid Integrity Treatment

Mixed results were observed when testing the impact of the PtCl_4_ treatment on HuNoV and MNV. Indeed, when the PtCl_4_ capsid integrity treatment was added to the ISO 15216-1:2017 extraction process, the non-heat-treated HuNoV RT-qPCR and long RT-qPCR recovery yields were significantly decreased to 31 ± 6% and 40 ± 6% (n = 6), respectively (*p* < 0.05). Meanwhile, the non-heat-treated MNV recovery was increased when detected by long RT-qPCR to 88 ± 20% (n = 6), while the RT-qPCR result variation was not significant. Nevertheless, the PtCl_4_ had a significant impact on the heat-treated HuNoV recovery yields as estimated by the RT-qPCR and long RT-qPCR with recovery yields at 6 ± 3% (n = 6) and 0.1 ± 0.2% (n = 6), respectively (*p* < 0.05). While six heat-treated HuNoV samples tested positive by RT-qPCR, only one tested positive by long RT-qPCR. In the meantime, all heat-treated MNV samples that were recovered following the PtCl_4_ treatment were detected by both methods. In comparison to HuNoV, the heat-treated MNV RT-qPCR and long RT-qPCR recovery yields were relatively higher at 22 ± 12% (n = 6) and 11 ± 6% (n = 6), respectively (*p* < 0.05).

#### 3.3.4. RNase Capsid Integrity Treatment

The addition of the RNase capsid integrity treatment into the ISO 15216-1:2017 extraction process did not impact the non-heat-treated HuNoV recovery results when compared to the control. However, it had an impact on non-heat-treated MNV RT-qPCR recovery results which were reduced to 26.9 ± 5.7% (n = 12) (*p* < 0.05). When the heat-treated viruses spiked on lettuce were treated with RNase in the extraction process, the HuNoV RT-qPCR and long RT-qPCR recovery yields decreased to 0.5 ± 0.2% and 0.3 ± 0.3% (n = 17), respectively (*p* < 0.05). Similar observations were made with heat-treated MNV using the RT-qPCR and the long RT-qPCR, as recovery yields decreased to 1.5 ± 0.9% and 3 ± 5% (n = 17), respectively (*p* < 0.05).

## 4. Discussion

The early detection of outbreaks and identification of contaminated food products are crucial in reducing the health burdens of foodborne outbreaks. However, even though food recalls as well as manufacturer and distributor voluntary product removal are effective measures to prevent foodborne illnesses, these measures are also associated with significant direct and indirect costs [31]. Yet, the early detection of contaminated food samples is based on RT-qPCR methodologies and the interpretation of results could have some limitations, as we observed in this study with the relatively high heat-treated virus recovery rates. Heat-treated viruses can be detected by RT-qPCR because the heat inactivation process alters the capsid integrity, while the viral genome remains relatively unaltered or fragmented. Moreover, RNA from heat inactivated viruses inoculated on food, water and surfaces can remain detectable by RT-qPCR for several days [32]. As a result, while the RT-qPCR detection results could be used as a virus contamination indicator, they are not always a proper indicator of their infectivity. Alternative approaches to evaluate the virus integrity could provide valuable information to assess the risk associated with the RT-qPCR results [13,33].

An ideal virus integrity detection method should have a high sensitivity while minimizing false-positive results. For HuNoV, lacking a sensitive culture system, the proportion of infectious virus could be overestimated based on the genomic copy levels. In addition to the viral capsid integrity, several factors influence the correlation between the viral genomic copy levels and virus integrity measurements. For instance, the presence of incomplete virus particles, degraded genomic RNA, and defective interfering or non-infective virus particles have an impact on this correlation [34,35]. Even when the viruses are intact, virus aggregation has a significant impact on infectivity quantification [13]. As a result, the proportion of intact viruses in the samples tested are unknown [18]. The interpretation of the impact of the capsid integrity treatments is limited by these factors.

Nevertheless, we estimated that in our model, no intact virus should be detected after the heat treatment. Heat treatment is a classical approach to release and extract RNA from virus samples [36]. However, only a few studies have investigated the heat inactivation temperatures of HuNoV based on its growth on culture cells. HuNoV GII.4 heat-treated for 1 min at 90 °C and spiked on freshwater clams could not be cultivated on human intestinal enteroids, while they remained detectable by RT-qPCR with a recovery rate of 28% [37]. Using human enteroids, Ettayebi, et al. [38] reported the complete heat inactivation of GII.3 and GII.4 HuNoVs after a heat treatment of 15 min at 60 °C. Most research groups have studied heat-treated noroviruses inactivation using surrogates [6,13,39]. Exposure to 80 °C for 150 s was sufficient to inactivate MNV using plaque forming or transfection assays [27]. Other groups concluded that at least a 6 log reduction of MNV infectivity could be achieved after 8 s to 16 s treatment at 80 °C [40,41]. However, these various studies were conducted at high virus titer (10^6^ to 10^10^ gEq) that do not reflect the concentrations reported in contaminated food [30,42,43]. Since in the current study the inoculated virus levels (<10^5^) allowed the detection of a maximum of 3 log reduction, only free RNA including degraded RNA, and no intact HuNoV or MNV, should be detected following the selected heat treatment. Although providing virus integrity information along with the HuNoV routine testing results is the main objective of this study, further research to assess the impact of the integrity treatments on the surrogate virus MNV infectivity in culture, using higher inoculum titers, might provide additional insight.

The PMAxx capsid integrity treatment combined with the long RT-qPCR detection method was the only treatment to eliminate completely both heat-treated HuNoV and MNV recovery. The procedure was relatively easily integrated into the ISO protocol, although in our hands, it added about 1 h to the extraction process. This was not negligible considering the ISO 15216-1:2017 processing time (~8 h). The efficacy of the intercalating dye approach depends on parameters associated with the viability marker such as the type of dye, the concentration and the incubation condition [14]. The addition of the non-ionic surfactants, such as Triton X-100, was reported to enhance the PMAxx capacity to enter partially or completely ruptured capsids, where it can covalently bind to nucleic acids upon photoactivation [44,45]. The group of Randazzo, et al. [45] reported that the RT-qPCR signal of heat-treated HuNoV spiked on lettuce was reduced by 1 to >3 logs by the PMAxx with triton treatment, which is similar to the levels we observed with HuNoV (>3 log). The group of Terio, et al. [46] also successfully applied this capsid integrity treatment to ready-to-eat vegetables, including mixed salads extracted using the ISO 15216-2:2017 protocol. They estimated that the addition of this capsid integrity treatment provided a better evaluation of the native viral particles that are dangerous for public health. However, they also reported a 4.1 Ct reduction of the non-heat-treated HuNoV signal when using the PMAxx treatment. We did not observe such a reduction with the non-heat-treated HuNoV RT-qPCR in terms of recovery rates, but a 0.5 log reduction was noticed with MNV recovery rates. Other groups have reported that the target virus, the virus inactivation treatment and the matrix being tested might influence the efficacy of the PMAxx approach [11,47]. Since the proportion of infectious viruses in the original samples is unknown, the interpretation of the non-heat-treated virus reduction remains inconclusive. It suggests that the impact of the treatment on other genotypes should also be investigated. Based on the long RT-qPCR results, the PMAxx treatment impact on non-heat-treated HuNoV integrity varies with the detection method and might not be negligible. Nevertheless, in our model, the PMAxx RT-qPCR method provided a good differentiation between non-heat-treated and heat-treated HuNoV GII.

Alternatively, platinum compounds such as platinum (IV) chloride (PtCl_4_) are known to be chelated by nucleic acid ligands and were also reported to be an effective pre-treatment to improve the correlation between RT-qPCR results and virus integrity [15]. They were easily integrated into the ISO protocol, adding only 10 min to the overall process. Moreover, they represented an inexpensive alternative to monoazide dyes [15,48]. Using RT-qPCR, Fraise, et al. [15] reported that the recovery rates of HuNoV GII and MNV heat-treated at 80 °C for 5 min were reduced by approximately 2.5 log following a 2.5 mM PtCl_4_ treatment. In contrast, others have observed no significant differences when comparing HuNoV GI and GII quantification with and without PtCl_4_ pre-treatment [49]. In the present study, the PtCl_4_ treatment reduced the recovery of both heat-treated viruses when evaluated using the short RT-qPCR, although to a lower extent than the other capsid treatments. However, it also significantly reduced the recovery of untreated HuNoV. It suggests that the incubation conditions of this viability marker, or its inactivation, were not optimal. Fraise, et al. [15] reported that the titer of only one of seventeen native HuNoV GII strains tested was reduced by PtCl_4_ treatment. The impact on non-heat-treated HuNoV virus, as well as the lower reduction in the heat-treated RT-qPCR signal compared to the other capsid integrity treatments, indicated that the PtCl_4_ protocol requires further improvement.

Several studies have shown that the RNase treatments, either RNase A or RNase ONE^TM^, were effective approaches to allow the quantification of intact RNA viruses by RT-qPCR [25,41,50,51]. For instance, when HuNoV were heat-treated at 76 °C for 2 min, Topping, et al. [25] reported that no HuNoV could be detected by RT-qPCR following the RNase ONE^TM^ treatment. They proposed a linear model where the capsid is altered, and the complete RNA is simultaneously exposed to RNase and degraded [25]. On the other hand, when heat treated at 80 °C for 8 s, Bartsch, et al. [41] reported that no infectious MNV could be detected, while the HuNoV RNA, following the integrated RNase A treatment, was still detected and decreased by 3.5 log as estimated by RT-qPCR. Whether the efficiency of the different capsid integrity approaches could be suboptimal with rapid or low heat treatment remains to be defined. On the other hand, an excess of RNase A was reported to decrease undamaged MNV recovery from the sample eluted using the ISO protocol [50]. In the present study, the RNase ONE^TM^ treatment was not associated with a lower recovery of non-heat-treated HuNoV while the MNV RT-qPCR signal was reduced. Still, the RNase RT-qPCR approach effectively differentiated the non-heat-treated from the heat inactivated HuNoV GII.

The impact of the residual RNase from the food matrix as well as the clinical samples (HuNoV) and culture cell extract (MNV) on the recovery results is probably not negligible. The inocula are not free of RNase. As proposed by Topping, et al. [25], the absence of differences between RNase and control non-heat-treated HuNoV groups without matrix demonstrates that no free-viral RNA was present in the inocula. Conversely, the presence of residual RNase in the non-heat-treated inocula also suggests that the HuNoV genomes tested in the study were relatively protected from enzymatic degradation. Since the MNV inoculum was produced in culture, some remaining free-viral MNV RNA could explain the differences in recovery rates between RNase and control as detected by RT-qPCR. The fact that no significant differences were observed using the long RT-qPCR between those non-heat-treated treatments might suggest that this difference resulted mainly from partially degraded, free MNV RNA. Further tests to define the residual RNase activities with or without matrices could clarify some aspects of these hypotheses.

Once heated, the viral genomes become accessible to RNases that are present in and on the surface of the food and are gradually fragmented and eliminated over time [20,25,52]. However, in this study, the residual RNase activity was not sufficient by itself to reduce or eliminate the false-positive signal from the heat-treated control since their recovery rates, as detected by RT-qPCR, were still relatively high (19 to 44%). It confirms that unless the food matrices are exposed to RNase degradation conditions for a relatively prolonged time, current short RT-qPCR detection protocols would not differentiate free-RNA from intact encapsidated viral RNA. As described above, capsid treatments such as RNase and PMAxx, as well as to a lesser extent PtCl_4_, are effective approaches to reduce the free-RNA RT-qPCR signal.

On the other hand, the impact of the long RT-qPCR detection approach, when used alone, on the recovery rates of both heat-treated HuNoV and MNV spiked on the matrix was also quite significant. When spiked on lettuce, the recovery rates estimated by long RT-qPCR of heat-treated HuNoV and MNV were approximately 1 and 0.5 log lower than with the short RT-qPCR, respectively. This level of reduction probably results from the residual RNase activity, since the genome integrity is not directly impacted by the heating process at the level used in our model [27]. This residual RNase activity could originate from the food surface or the extraction process. Similar conclusions were made by Wolf, et al. [20]. Following the complete heat inactivation of MNV at 72 °C, at 5 min they reported a 0.5 log difference between short and long RT-qPCR. The longer the distance between the RT-priming site and the PCR amplification sites, the less likely fragment RNA would be detected [20]. They proposed that the MNV recovery reduction was due to RNA cleavage by free RNases subsequent to the disruption of the virus capsid. Testing heat inactivated MNV [53] indicated that there could be up to a 3 log difference in genome copy estimate depending upon which region is targeted for the detection and whether or not a long-range RT-qPCR is used. They also reported that MNV genome copy detected following inactivation increased when using primer sets for long RT-qPCR that were closer to the 3′end. Indeed, the probability that RNase would degrade the free target viral RNA increases with the length of the target region. These observations suggest that the longer the target RNA fragment sizes are, the more their concentrations should correlate to the undamaged viruses.

Still, there is some limitation to the use of long genome fragment as a virus integrity indicator. First, the long RT-qPCR detection approach could be associated with a decrease in sensitivity [19,53]. Indeed, we observed an overall increase of the HuNoV inoculum cycle thresholds (3.6 Ct) using the long RT-qPCR compared to the short RT-qPCR. This increase is explained in part by the fraction of the RT volumes (1/4) used with the two steps long RT-qPCR compared with the one step RT-qPCR. This difference is equivalent to 2 Ct, and could be avoided by increasing the RT volume tested using multiple qPCRs that involve additional times and costs [30]. Part of the difference, equivalent to 1.4 Ct, was also associated with the ratio of long to short viral genome fragments present in the inoculum. Thus, the decrease in sensitivity might impact the detection of fragment RNA but not intact virions. Nevertheless, while these factors could explain most of the observed difference, we cannot exclude a bias since we used different RT-qPCR conditions. The true ability of long range RT-qPCR to distinguish infectious and non-infectious HuNoV remains to be tested against culture systems in order to measure the infectivity or lack thereof.

The second limitation to the use of a long genome fragment is the heat-treated model itself. It reflects, only in part, the range of conditions that could impact the virus integrity. There could be a range of impacts below some temperature thresholds, i.e., temperatures where the virus capsid partially loses some integrity while degraded and long RNA fragments still remain protected. For instance, Li, et al. [18] reported only a limited reduction (0.3 log) in MNV using either long RT-qPCR or short RT-qPCR approaches after a heat treatment of 60 °C for 30 min, while MNV-1 infectivity was reduced to a non-detectable level (~4 log PFU reduction). In such a case, there is no clear advantage to using the long RT-qPCR or further optimisation is required. In contrast, it is expected that chlorine treatment will degrade both the capsid and the genome, while the UV could have a greater impact on the genome first [17,54,55]. In those cases, targeting the genome rather than the capsid integrity might be more informative regarding virus integrity. In any case, it is worth documenting and linking the production, processing and distribution chain of the target food product to select the more appropriate detection methodology and to assess if reducing the false positive rate provides any advantages.

Despite these issues, using the two step long RT-qPCR and similar long-range RT PCR alone has several advantages. Compared to the other capsid integrity treatments, these detection methods do not require a modification of the extraction protocols. They could be applied to specific cases such as to retest or to confirm presumptive positive RT-qPCR samples. For a food safety laboratory testing thousands of samples annually, it definitely does not have the same negative impact as the capsid integrity treatments in terms of processing times and costs. Furthermore, some long-range RT approaches can be combined with PCR and genomic sequencing tools to provide information for traceback investigations, as was demonstrated with contaminated frozen raspberries [30]. Following this concept, long-range RT PCR, and even whole genome sequencing, have additional value since they could be used as a confirmation tool to link outbreak events and provide information on virus integrity.

## 5. Conclusions

Current noroviruses RT-qPCR detection protocols do not differentiate infectious from non-infectious viruses. False positives associated with degraded virus or naked RNA following a heat treatment process are reduced by using virus PMAxx, RNase or PtCl_4_ capsid integrity treatments. On the other hand, longer genome detection, using long RT-qPCR, can also be used as a proxy to evaluate viral integrity when HuNoV are exposed to RNases, as we demonstrated with spiked lettuce. The efficiency of these genomic and capsid treatments could vary with the target virus. Nevertheless, these protocols improve the virus integrity assessment and can be effectively included in foodborne virus extraction and detection processes.

## Figures and Tables

**Figure 1 foods-12-00826-f001:**
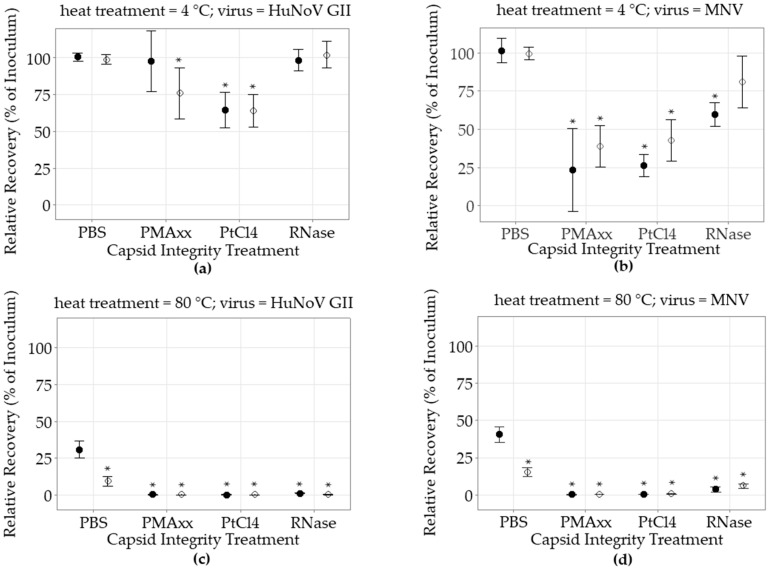
Impact of capsid integrity treatment on heat-treated norovirus recovery without food matrices. Viruses were heat-treated for 5 min or left untreated; (**a**) non heat treatment (4 °C) HuNoV GII; (**b**) non heat treatment (4 °C) MNV; (**c**) heat treatment (80 °C) HuNoV GII; (**d**) heat treatment (80 °C) MNV. Following heat treatment, the viruses were treated with either PBS, PMAxx, PtCl_4_, or RNase, and then the RNA was extracted and detected. Relative recovery results are expressed as percentage of the spiked inoculum as estimated by short RT-qPCR (●) or long RT-qPCR (○) (mean ± 95% CI). The Dunnett’s multiple comparison test was used to identify significant differences (*) compared to the short RT-qPCR recovery of the virus without capsid treatment (PBS) (*p* < 0.05).

**Figure 2 foods-12-00826-f002:**
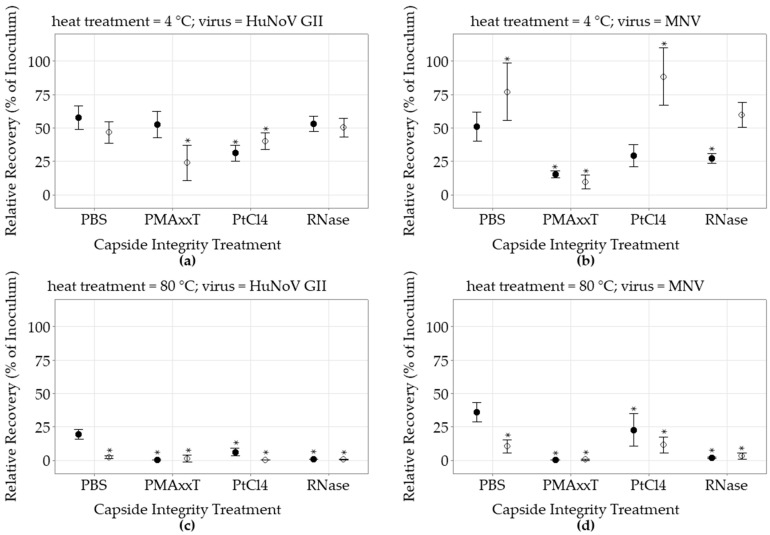
Impact of capsid integrity treatment on heat-treated norovirus recovery following ISO 15216-1:2017 extraction from spiked lettuce. Viruses were heat treated for 5 min or left untreated; (**a**) non heat treatment (4 °C) HuNoV GII; (**b**) non heat treatment (4 °C) MNV; (**c**) heat treatment (80 °C) HuNoV GII; (**d**) heat treatment (80 °C) MNV. Following heat treatment, the viruses were spiked on lettuce then eluted and concentrated using the ISO 15216-1:2017 protocol, and treated with either PBS, PMAxx, PtCl_4_ or RNase before the RNA extraction and detection. Relative recovery results are expressed as percentage of the spiked inoculum, as estimated by short RT-qPCR (●) or long RT-qPCR (○) (mean ± 95% CI). The Dunnett’s multiple comparison test was used to identify significant differences (*) compared to the short RT-qPCR recovery of the virus without capsid treatment (PBS) (*p* < 0.05).

## Data Availability

The datasets generated during and/or analysed during the current study are available from the corresponding author on reasonable request.

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
