# Peer review of "Impact of Capsid and Genomic Integrity Tests on Norovirus Extraction Recovery Rates"

_foods, 2023, doi:10.3390/foods12040826_

Round 1

Reviewer 1 Report

This paper documents the application of three capsid integrity assessment approaches combined with a genomic integrity approach on heat treated and non-heat treated human norovirus (GII.4) and murine norovirus in a food (lettuce) and non-food matrix. A considerable amount of work was undertaken to implement the various approaches with two different noroviruses and matrix effects and offers some solutions towards differentiating between infectious and non-infectious norovirus by RT-qPCR. This is commendable however the way the results are presented and discussed in some areas are not clear and would benefit from some modifications.

Specific comments:

1.       Line 54-55: Consider modifying this sentence to “…the detection of non-infectious virus by qPCR could be the equivalent of a false-positive result.” As a general note the authors should consider carefully how they use the term false-positive. False positives in PCR are generally associated with PCR contamination. The detection of non-infectious virus by PCR should not be considered as a false positive result. The very nature of PCR is designed to detect viral RNA whether it is infectious or not. This is undoubtedly a disadvantage of PCR detection of viral pathogens in food however qPCR remains an important tool for in foodborne virus risk management.

2.       Why wasn’t norovirus infectivity assessed using MNV cell culture? This could help put the results in more context and reveal the true impact of the pre-treatments and long range PCR methods. Its absence is a weakness of the study.

3.       Line 177: ~0.1 bp; should this be ~0.1 kb?

4.       A diagram to show the location of the primers on the human and murine norovirus genomes would be useful

5.       Section 2.9 Long to short viral genome fragments ratio – it might be useful to rephrase how the formula is interpreted. It is currently a very long sentence (lines 180 – 184) and could be more concise or broken down to enable clarity.

6.       It isn’t clear from the text as to what the significance of this ratio is or how it was applied to human norovirus for example. Further explanation would be helpful.

7.       For viral RNA recovery yields why were serial dilutions of the norovirus RNA used as the standard curve for the long RT qPCR instead of RNA transcripts? RNA transcripts were used for the RT-qPCR assay. Could the different approaches to slope calculation interfere with the recovery calculations?

8.       It would be useful to include a table of the log reductions of virus pre and post heat treatment. Reference is made in line 389-390 that a >3 log reduction was observed in the study post heat treatment with the inclusion of PMAxx in the detection protocol however this result isn’t clear from the figures presented.

9.       The inclusion of log reductions could shed light on how much of the virus integrity is revealed by (a) the genomic integrity assessment and (b) by the capsid integrity.

10.   Given the various treatments and two PCR approaches applied it is difficult to ascertain the number of replicates included in each experiment, what was carried out on individual samples, etc. A graphical representation or flow chart type figure would be extremely useful here.

11.   Two different RNA extraction approaches were used; one for lettuce extracts and a different kit was used for PBS samples. It is interesting as to why this approach was taken. Was there a particular reason as to why the NucliSens miniMag kit wasn’t used on the PBS samples?

12.   Line 304-307: the HuNoV short and long RT-qPCR recovery yields were significantly decreased, however it is not clear what they are being compared to in this instance. Better explanation on how the short and long range RT-PCR approaches are being compared would be helpful. Are they being compared to each other all the time or to is it how they perform in PBS versus spiked lettuce, this is not always clear. 

Author Response

Thank you for comments and recommendations regarding our manuscript "Impact of Capsid and Genomic Integrity Tests on Norovirus Extraction Recovery Rates".

Please see the attachment with our point-by-point response.

Reviewer 2 Report

Raymond et al describe a comparison of normal RT-qPCR with long RT qPCR to evaluate a capsid integrity of HuNoV and MNV that were treated with RNase, PMAxx and PtCl4. Briefly, the authors spiked the viruses, which were heat and non-heat treated, on lettuce and were subjected to extraction and concentration of the viruses by the ISO 15216-1:2017 protocols. Relative recovery rates of the viruses acquired by normal RT-qPCR and long RT qPCR were compared, showing a merit that long RT qPCR could be used as a proxy to evaluate viral integrity when HuNoV are exposed to food RNases.

  The reviewer thinks that the comparison of the RT-qPCRs with several treatments would be beneficial to develop methods to evaluate the infectivity of viruses. However, at first, this manuscript had to be completed before submission, and requires more description of novelty of this study. In this study, HuNoV and MNV were tested by RT-qPCR to evaluate infectivity. Although the reviewer agrees an adequacy of the test for HuNoV due to the difficulty of using enteroids, a propagation of MNV could be adopted for this study as direct evidence for virus integrity.

Line 29. Please unify singular or plural form of the abbreviations of viruses.

Line 146. Do you have control treated with UV in the absence of PMAxx?

Line 164. "synthesis cDNA" might be "synthesis of cDNA"?

Lines 445 and 455. "RNAse" should be "RNase".

This manuscript still includes past figures and references that must be deleted before submission.

Author Response

(The authors gave the same response as above.)

Reviewer 3 Report

Dear Authors, the article is well written, clear and well structurated. This study is interesting and relevant for the fields. I have only some minor revisions to suggest you:

Introduction:

References 1 and 4 they are a bit dated better to put them more up to date.

Line 46: Add a reference after a low infectious dose (approximately 18 to 1000 virus particles).

Line 54: Add a reference at the end of HPP

Line 84: In the scope it is not said that the experiments are done both on food matrices and not. Please add.

Line 263: Them change with then 

Line 506-507: check the sentence "In such a case, there is no clear advantage to using the long RT qPCR or the long RT qPCR needed optimisation"..... Long RT-qPCR appears twice. It's correct?

Author Response

(The authors gave the same response as above.)

Reviewer 4 Report

The manuscript by Raymond et al reported methods to estimate capsid and genomic integrity of human norovirus and MNV.  The paper was well written with proper conclusions.  The paper could be divided into two parts.   The first was PBS diluted sample and the second was eluted samples from lettuce.  I think it does not need the 2nd part included.  These two sets of samples basically are the same (washing off air-dried PBS-diluted samples with or without treatments from lettuce).  The concept of the paper should be well presented by the 1st part.  A minor comment is relative recovery (% of inoculation) used in Fig 1 and 2 have different meanings.  In fig 1., should be impact of three treatments on viral detection and Fig. 2 is related to recovery. The manuscript could be considered acceptable after a minor revision. 

Author Response

Thank you for comments and recommendations regarding our manuscript "Impact of Capsid and Genomic Integrity Tests on Norovirus Extraction Recovery Rates".

Reviewer' comments: The manuscript by Raymond et al reported methods to estimate capsid and genomic integrity of human norovirus and MNV.  The paper was well written with proper conclusions.  The paper could be divided into two parts.   The first was PBS diluted sample and the second was eluted samples from lettuce.  I think it does not need the 2nd part included.  These two sets of samples basically are the same (washing off air-dried PBS-diluted samples with or without treatments from lettuce).  The concept of the paper should be well presented by the 1st part.  A minor comment is relative recovery (% of inoculation) used in Fig 1 and 2 have different meanings.  In fig 1., should be impact of three treatments on viral detection and Fig. 2 is related to recovery. The manuscript could be considered acceptable after a minor revision.

Author's reply:

The two parts are different.

The first part target the impact of the three treatments on the viral detection, using either short RT-qPCR or long RT qPCR to evaluate it. However, in order to evaluate their practical application, the second part is even more important since we evaluated how these approaches integrated with the standard ISO15261 protocols used in testing laboratories. The capsid integrity treatments have an impact in the processing time when integrated. We think the long RT qPCR, even if less sensitive, could be used to provide virus integrity information along with the HuNoV routine testing results and facilitate the public health risk assessment process.

The meaning of relative recovery is the same between Fig 1.  and Fig 2. They are both the evaluation of recovery compared to the inoculum (Line 192). However, in Fig 1 the inoculum is the same as the control PBS since there is no matrices. We added the precision in a new Supplementary Figure SF1 that describes the two experiments workflow.

Regards,

Philippe Raymond Phd.

Round 2

Reviewer 2 Report

This reviewer agrees the reviewer's response to my comments and suggests accepting after a bit minor correction as below.

1. The reverse primer for HuNoV should be "COG2R" not "CO2R" and "COG2".

2. Please check if you need to correct "PMaxx" and "PMaxxT" in the Tables ST2 and ST3 to "PMAxx".

Author Response

 Thank you for reviewing our manuscript.

 We made the recommended minor corrections to the Supplementary Figure SF2 Primer and probe locations, Table ST1 and Table ST2.

Regards,

Philippe Raymond, PhD